# Comprehensive characterization of the antibody responses to SARS-CoV-2 Spike protein finds additional vaccine-induced epitopes beyond those for mild infection

Meghan E Garrett[1†], Jared G Galloway[2†], Caitlin Wolf[3], Jennifer K Logue[3], Nicholas Franko[3], Helen Y Chu[3], Frederick A Matsen IV[4]*, Julie M Overbaugh[1]*

[1]Division of Human Biology, Fred Hutchinson Cancer Research Center, Seattle, United States; [2]Division of Public Health Sciences, Fred Hutchinson Cancer Research Center, Seattle, United States; [3]Department of Medicine, University of Washington, Seattle, United States; [4]Computational Biology Program, Fred Hutchinson Cancer Research Center, Seattle, United States

*For correspondence:
matsen@fredhutch.org (FAM);
joverbau@fredhutch.org (JMO)

†These authors contributed equally to this work

## Abstract

**Background:** Control of the COVID-19 pandemic will rely on SARS-CoV-2 vaccine-elicited antibodies to protect against emerging and future variants; an understanding of the unique features of the humoral responses to infection and vaccination, including different vaccine platforms, is needed to achieve this goal.

**Methods:** The epitopes and pathways of escape for Spike-specific antibodies in individuals with diverse infection and vaccination history were profiled using Phage-DMS. Principal component analysis was performed to identify regions of antibody binding along the Spike protein that differentiate the samples from one another. Within these epitope regions, we determined potential sites of escape by comparing antibody binding of peptides containing wild-type residues versus peptides containing a mutant residue.

**Results:** Individuals with mild infection had antibodies that bound to epitopes in the S2 subunit within the fusion peptide and heptad-repeat regions, whereas vaccinated individuals had antibodies that additionally bound to epitopes in the N- and C-terminal domains of the S1 subunit, a pattern that was also observed in individuals with severe disease due to infection. Epitope binding appeared to change over time after vaccination, but other covariates such as mRNA vaccine dose, mRNA vaccine type, and age did not affect antibody binding to these epitopes. Vaccination induced a relatively uniform escape profile across individuals for some epitopes, whereas there was much more variation in escape pathways in mildly infected individuals. In the case of antibodies targeting the fusion peptide region, which was a common response to both infection and vaccination, the escape profile after infection was not altered by subsequent vaccination.

**Conclusions:** The finding that SARS-CoV-2 mRNA vaccination resulted in binding to additional epitopes beyond what was seen after infection suggests that protection could vary depending on the route of exposure to Spike antigen. The relatively conserved escape pathways to vaccine-induced antibodies relative to infection-induced antibodies suggests that if escape variants emerge they may be readily selected for across vaccinated individuals. Given that the majority of people will be first exposed to Spike via vaccination and not infection, this work has implications for predicting the selection of immune escape variants at a population level.

**Funding:** This work was supported by NIH grants AI138709 (PI JMO) and AI146028 (PI FAM). JMO received support as the Endowed Chair for Graduate Education (FHCRC). The research of FAM was

supported in part by a Faculty Scholar grant from the Howard Hughes Medical Institute and the Simons Foundation. Scientific Computing Infrastructure at Fred Hutch was funded by ORIP grant S10OD028685.

## Editor's evaluation

In this paper, antibody responses to SARS-CoV-2 Spike protein were investigated in mRNA vaccinated, SARS-CoV-2 infected with mild or severe COVID-19 and vaccinated individuals with prior infection. Key findings of the study are that severe COVID-19 and vaccinated individuals had higher binding to Spike protein regions NTD, CTD, and SH-H compared to individuals with mild COVID-19; while mild COVID-19 infections had higher binding to FP than vaccinated or severe COVID-19 individuals. Individuals with or without prior infection were not different and that covariates did not appear to impact the antibody recognition profiles. The authors identified potential escape pathways in these epitope regions, some of which differed between vaccination and infection or drifted over time. The authors acknowledge that this approach is limited to linear epitopes and does not include receptor-binding domain epitopes. The study provides novel insight into the major epitope regions targeted by polyclonal antibodies elicited by vaccination vs. infection, as well as potential pathways for the virus to escape recognition.

## Introduction

The future of the COVID-19 pandemic will be determined in large part by the ability of vaccine-elicited immunity to protect against current and future variants of the SARS-CoV-2 virus. Several vaccines have now been approved for use in multiple countries, including two that are based on mRNA technology: BNT162b2 (Pfizer/BioNTech) and mRNA-1273 (Moderna). In the United States, over half of adults are now vaccinated against SARS-CoV-2, the majority of whom have received one of these mRNA vaccines. While these vaccines have been shown to effectively guard against infection, severe disease, and death related to SARS-CoV-2 (*Polack et al., 2020*; *Keehner et al., 2021*; *Amit et al., 2021*; *Angel et al., 2021*; *Thompson et al., 2021*; *Haas et al., 2021*; *Baden et al., 2021*), less is known about how effective they will be against emerging and future variants. One example is the recent surge of the Delta variant coupled with reports of reduced potency of vaccine-elicited antibodies against this variant, highlighting this concerning ongoing dynamic (*Planas et al., 2021*; *Lopez Bernal et al., 2021*) – a situation that is playing out in an even more significant way with the Omicron variant. Evidence from related endemic coronaviruses indicates that evolution in the Spike protein results in escape from neutralizing antibodies elicited by prior infection (*Eguia et al., 2021*), potentially contributing to why endemic coronaviruses can reinfect the same host (*Edridge et al., 2020*; *Hendley et al., 1972*; *Schmidt et al., 1986*). Without immunity that is robust in the face of antigenic drift, continual updates of the vaccine to combat new SARS-CoV-2 variants will likely be necessary to provide optimal protection against symptomatic infection.

Prior infection with SARS-CoV-2 also provides some immunity against subsequent reinfection, and several studies have characterized the epitopes targeted by convalescent sera (*Hanrath et al., 2021*; *Greaney et al., 2021a*; *Shrock et al., 2020*; *Li et al., 2021*; *Stoddard et al., 2021*). It is currently unknown whether SARS-CoV-2 infection and vaccination result in antibodies that bind to similar epitopes, an important point to consider given that most people have acquired antibodies through immunization and not infection. The Spike protein encoded by the mRNA in both SARS-CoV-2 vaccines is stabilized in the prefusion conformation by addition of two proline substitutions (*Corbett et al., 2020*). This change in sequence and fixed conformation of the Spike protein could result in altered antibody targeting when compared to antibodies elicited during infection, where Spike undergoes several conformational changes. It is also possible that differences in antibody specificity could be due to the amount of antigen or type of immune response stimulated in the context of infection versus vaccination. We know that vaccines drive higher neutralization titers and more Spike binding IgG antibodies than infection (*Goel et al., 2021*; *Prendecki et al., 2021*; *Edara et al., 2021*), indicating some differences in the B cell response compared to infection. A recent study showed that antibodies against the receptor binding domain (RBD) of Spike differ between infected and vaccinated

**eLife digest** When SARS-CoV-2 – the virus that causes COVID-19 – infects our bodies, our immune system reacts by producing small molecules called antibodies that stick to a part of the virus called the spike protein. Vaccines are thought to work by triggering the production of similar antibodies without causing disease. Some of the most effective antibodies against SARS-CoV-2 bind a specific area of the spike protein called the 'receptor binding domain' or RBD.

When SARS-CoV-2 evolves it creates a challenge for our immune system: mutations, which are changes in the virus's genetic code, can alter the shape of its spike protein, meaning that existing antibodies may no longer bind to it as effectively. This lowers the protection offered by past infection or vaccination, which makes it harder to tackle the pandemic.

As it stands, it is not clear which mutations to the virus's genetic code can affect antibody binding, especially to portions outside the RBD. To complicate things further, the antibodies people produce in response to mild infection, severe infection, and vaccination, while somewhat overlapping, exhibit some differences. Studying these differences could help minimize emergence of mutations that allow the virus to 'escape' the antibody response.

A phage display library is a laboratory technique in which phages (viruses that infect bacteria) are used as a 'repository' for DNA fragments that code for a specific protein. The phages can then produce the protein (or fragments of it), and if the protein fragments bind to a target, it can be easily detected. Garrett, Galloway et al. exploited this technique to study how different portions of the SARS-CoV-2 spike protein were bound by antibodies. They made a phage library in which each phage encoded a portion of the spike protein with different mutations, and then exposed the different versions of the protein to antibodies from people who had experienced prior infection, vaccination, or both.

The experiment showed that antibodies produced during severe infection or after vaccination bound to similar parts of the spike protein, while antibodies from people who had experienced mild infection targeted fewer areas. Garrett, Galloway et al. also found that mutations that affected the binding of antibodies produced after vaccination were more consistent than mutations that interfered with antibodies produced during infection.

While these results show which mutations are most likely to help the virus escape existing antibodies, this does not mean that the virus will necessarily evolve in that direction. Indeed, some of the mutations may be impossible for the virus to acquire because they interfere with the virus's ability to spread. Further studies could focus on revealing which of the mutations detected by Garrett, Galloway et al. are most likely to occur, to guide vaccine development in that direction. To help with this, Garrett, Galloway et al. have made the data accessible to other scientists and the public using a web tool.

individuals; they are generally less sensitive to mutation and bind more broadly across the domain in the context of vaccination as compared to infection (*Greaney et al., 2021b*).

Although the majority of the serum binding response in SARS-CoV-2-infected and -vaccinated people is directed towards regions of the protein outside of the RBD epitopes (*Greaney et al., 2021a*; *Greaney et al., 2021b*; *Garrett et al., 2021*; *Piccoli et al., 2020*), few studies have examined the prevalence and escape pathways of these antibodies, especially in the setting of vaccination. Antibodies to linear epitopes in the S2 domain of Spike overlapping the fusion peptide (FP), and in the stem helix region just upstream of heptad repeat 2 (SH-H) region are found in serum from COVID-19 patients, and some studies suggest that these antibodies may be neutralizing (*Poh et al., 2020*; *Li et al., 2020*). These non-RBD responses may also be important contributors to non-neutralizing antibody activities, which have been associated with protection and therapeutic benefit in experimental SARS-CoV-2 models and with vaccine protection (*Schäfer et al., 2021*; *Tauzin et al., 2021*; *Winkler, 2021*; *Ullah et al., 2021*). Importantly, these epitopes lie in more conserved regions of Spike than RBD where functional constraints on variation may counter the selective pressure for viral escape.

To compare antibody immunity elicited by SARS-CoV-2 infection and vaccination, we used a high-resolution Spike-specific deep mutational scanning phage display library to profile the epitopes and sites of escape for serum antibodies from people who had been infected, vaccinated, or a combination

of both. This approach, called Phage-DMS, identified four non-RBD antibody binding epitopes across all samples: the FP and SH-H region in the S2 subunit, and the N-terminal domain (NTD) and C-terminal domain (CTD) in the S1 subunit of Spike. Antibodies to NTD and CTD were uniquely present in the setting of mRNA vaccination or severe infection, but mostly absent in mild COVID-19 cases. In vaccinated individuals, the magnitude of the response varied over time both to the CTD and SH-H epitopes. Other covariates, such as age, dose, and vaccine type, had no significant differences in the binding profiles observed. Of particular relevance to protection against emerging variants, infection and vaccination appear to shape the pathways of escape differently in different epitopes. In the FP epitope, which is a dominant response after infection, the escape pathway was maintained after subsequent vaccination; in the SH-H epitope, infection resulted in antibodies with diverse pathways of escape, whereas vaccination induced a highly uniform escape profile across individuals. Overall, these findings indicate that vaccination induced a broader antibody response across the Spike protein but induced a singular antibody response at the SH-H epitope, which could favor variants that emerge with these mutations.

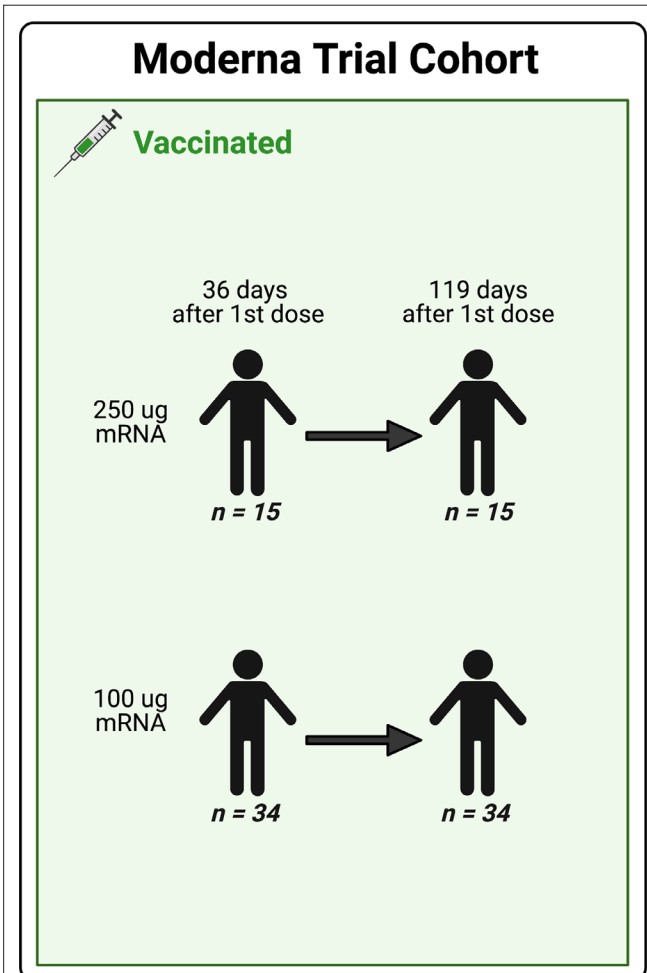

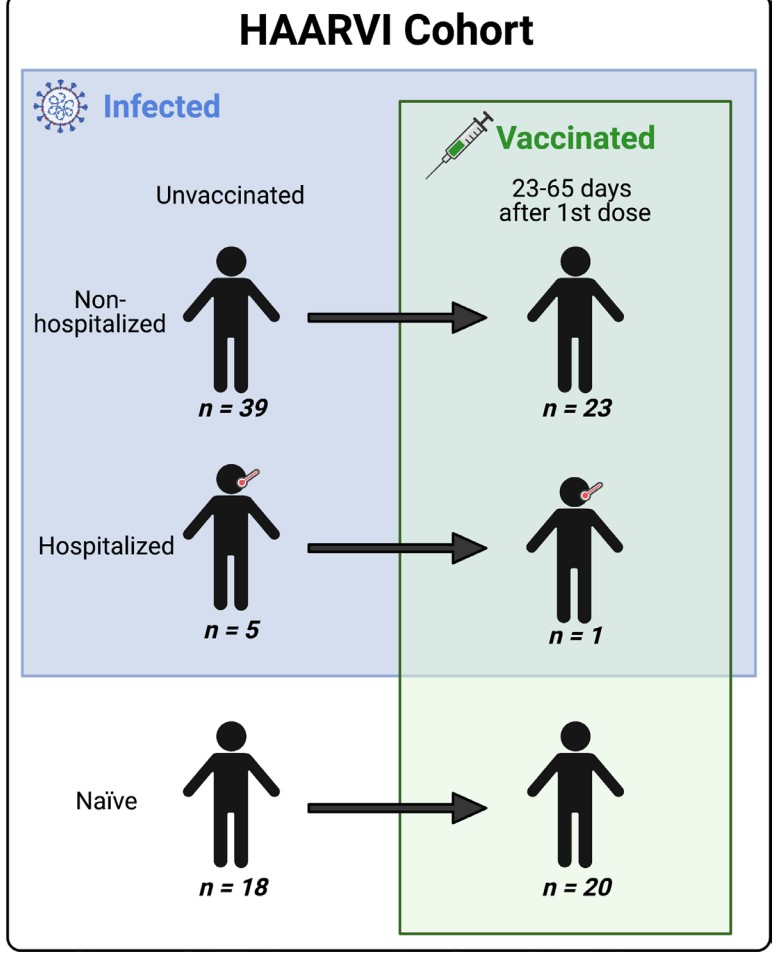

**Figure 1.** A schematic of sample cohorts. Characteristics of individual participants sampled as part of the Moderna Trial Cohort (left) or the Hospitalized or Ambulatory Adults with Respiratory Viral Infections (HAARVI) Cohort (right). Sample sizes of unique individuals in each group are designated below each figure.

## Results

### Samples from individuals with varying SARS-CoV-2 infection and mRNA vaccination histories profiled using high-resolution Spike Phage-DMS library

We collected serum samples from two cohorts, termed the Moderna Trial Cohort and the Hospitalized or Ambulatory Adults with Respiratory Viral Infections (HAARVI) Cohort (*Garrett et al., 2021*; *Jackson et al., 2020*). The Moderna Trial Cohort were participants in a Phase I trial and consisted of 49 individuals, 34 who received the 100 μg dose of mRNA-1273 (Moderna) and 15 who received the 250 μg dose. Serum samples were taken at days 36 and 119 post first dose (7 and 90 days post second dose, respectively; *Jackson et al., 2020*). The HAARVI Cohort included 64 individuals, 44 who had confirmed SARS-CoV-2 infection and 20 who had no reported infection; among this group, 44 were also vaccinated. Those with infection history were stratified by severity based on hospitalization status (39 nonhospitalized/mild vs. 5 hospitalized/severe), and serum was sampled at timepoints ranging from 8 to 309 days post symptom onset. Of these 44 individuals, 24 were also sampled after vaccination with two doses of either mRNA-1273 (Moderna, n = 8) or BNT162b2 (Pfizer/BioNTech, n = 15), with 23 from the nonhospitalized group and one from the hospitalized group. All 20 SARS-CoV-2-naïve individuals were sampled post vaccination, with 18 having an additional sample taken pre vaccination (0–98 days). Post-vaccination timepoints for all naïve and convalescent individuals ranged from 23 to 65 days after the first dose (5–42 days after the second dose, respectively). *Figure 1* provides an illustration of the two cohorts and their respective samples' infection and vaccination statuses. Additional details are available in *Supplementary file 1*.

We used a previously described Spike Phage-DMS library to profile the epitopes bound by serum antibodies in the samples described above (*Garrett et al., 2021*). This library consists of peptides displayed on the surface of T7 bacteriophage that are 31 amino acids long, tiling across the length of Spike in one amino acid increments. Peptides in the library correspond to the wild-type Wuhan Hu-1 Spike sequence as well as sequences that contain every possible single amino acid mutation at the central position of the peptide. Serum samples were screened with this library by performing immunoprecipitation (IP) followed by sequencing of the pool of phage enriched by the serum antibodies as previously described (*Garrett et al., 2021*; *Garrett et al., 2020*; *Mohan et al., 2018*).

### Serum antibodies bind to distinct epitopes in infected and vaccinated individuals

We first examined the wild-type peptides in the Spike Phage-DMS library that were enriched by each serum sample to determine the epitopes bound by antibodies in each sample from these cohorts (*Figure 2A*). The major targeted epitopes across all the cohorts were in the NTD, CTD, FP, and SH-H regions. Serum from nonvaccinated infected individuals who were not hospitalized mostly bound to immunodominant epitopes in the FP and SH-H, both of which are epitopes previously identified in infected individuals using Phage-DMS (*Garrett et al., 2021*). Samples from hospitalized/severe COVID-19 cases and vaccinated individuals also bound to the FP and SH-H regions, but additionally bound to epitopes within the NTD and CTD regions. In naïve serum samples, there were antibodies that occasionally bound to the FP and SH-H peptides. These findings likely reflect that some individuals have preexisting cross-reactive antibodies that bind to these conserved regions between SARS-CoV-2 and endemic coronaviruses, as suggested by previous studies (*Shrock et al., 2020*; *Stoddard et al., 2021*).

A principal component analysis (PCA) was used to further investigate differences between the infected and/or vaccinated groups (*Figure 2—figure supplement 1*). This analysis indicated that binding to epitopes in the NTD, CTD, FP, and SH-H regions was the driving difference between samples (*Figure 2B*). To quantify differences in antibody binding between groups, for each sample we summed together the enrichment values within each identified epitope region and performed pairwise comparisons between nonhospitalized infected people and all other groups (*Figure 2C*). Most strikingly, we found nontrivial group differences in the magnitude of humoral responses to these major epitopes on the Spike protein. Specifically, antibodies from both hospitalized infected and vaccinated individuals had significantly higher binding to the NTD, CTD, and SH-H regions compared to nonhospitalized infected individuals. However, antibodies from nonhospitalized infected individuals

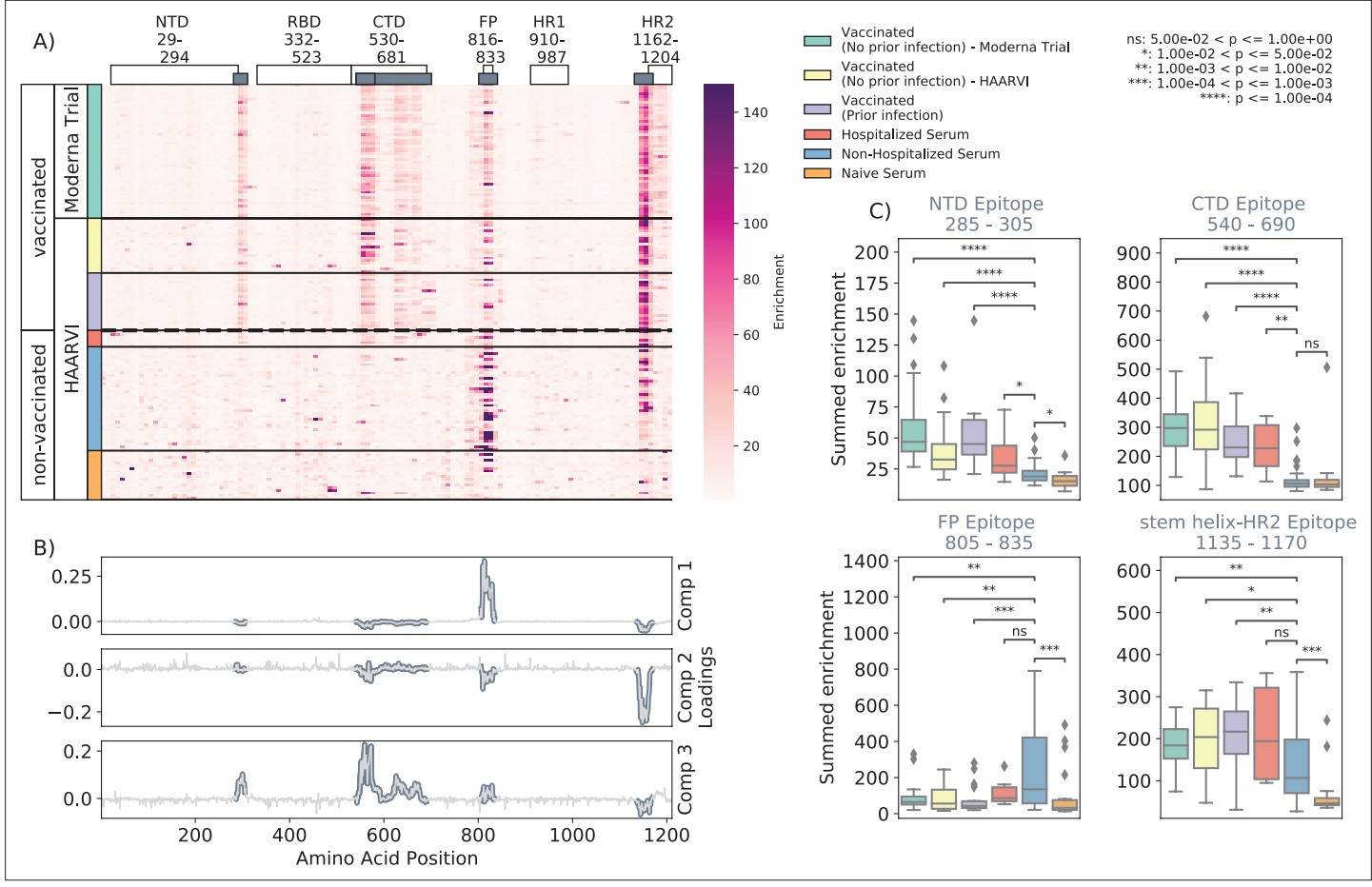

**Figure 2.** Enrichment of wild-type peptides by serum antibodies. (**A**) Heatmap with a sample in each row and groups of samples colored on the left. Columns represent peptide locations, with each square on the heatmap indicating the summed enrichment value within a 10-peptide interval. Darker purple indicates higher enrichment values, and values above 150 were capped. Transparent boxes above the heatmap annotate the Spike protein domains, while the smaller gray boxes indicate *epitope binding regions* defined in this analysis (**B**) The loading vectors from the principal component analysis with the four epitope sites highlighted; enrichments in each of these regions are summed together for subsequent analysis. (**C**) Boxplots describing the distribution of summed wild-type enrichment values for each sample within each of the four epitope sites, each named according to its associated protein domain. Color indicates the sample group. The bars between boxplots give statistical significance (p-value) tests using a Mann–Whitney–Wilcoxon test. All sample group comparisons with the nonhospitalized infected group were performed, and only significant values are shown.

The online version of this article includes the following figure supplement(s) for figure 2:

**Figure supplement 1.** Principal component analysis on wild-type enrichment features of all samples.

displayed significantly higher binding to the FP epitope than samples from hospitalized or vaccinated individuals. There was no significant difference in any epitope binding in these four regions between vaccinated samples with and without prior infection (p>0.05, Mann–Whitney–Wilcoxon [MWW]).

## Effect of age, dose, vaccine type, and timepoint on epitope binding

In order to determine if there were covariates that contributed to differences in antibody binding, we examined the effect of participant age, vaccine dose and type, and timepoint post infection or vaccination on binding to the four epitopes identified above (*Figure 3*). For samples in the Moderna Trial Cohort, there was significantly decreased binding to the CTD epitope and SH-H epitope (p=0.008, p=0.011, Wilcoxon rank-sum test with Bonferroni correction) at the later timepoint post first dose (day 119) compared to the earlier timepoint (day 36) (*Figure 3A*). To examine the effect of dosage, we compared 100 µg and 250 µg mRNA-1273 groups for those between the age of 18–55, as that was the only age group included for the 250 µg dose (*Figure 3B*). There was no significant difference by vaccine dosage for any of the four epitope regions (NTD, CTD, FP, or SH-H). Participant age was also

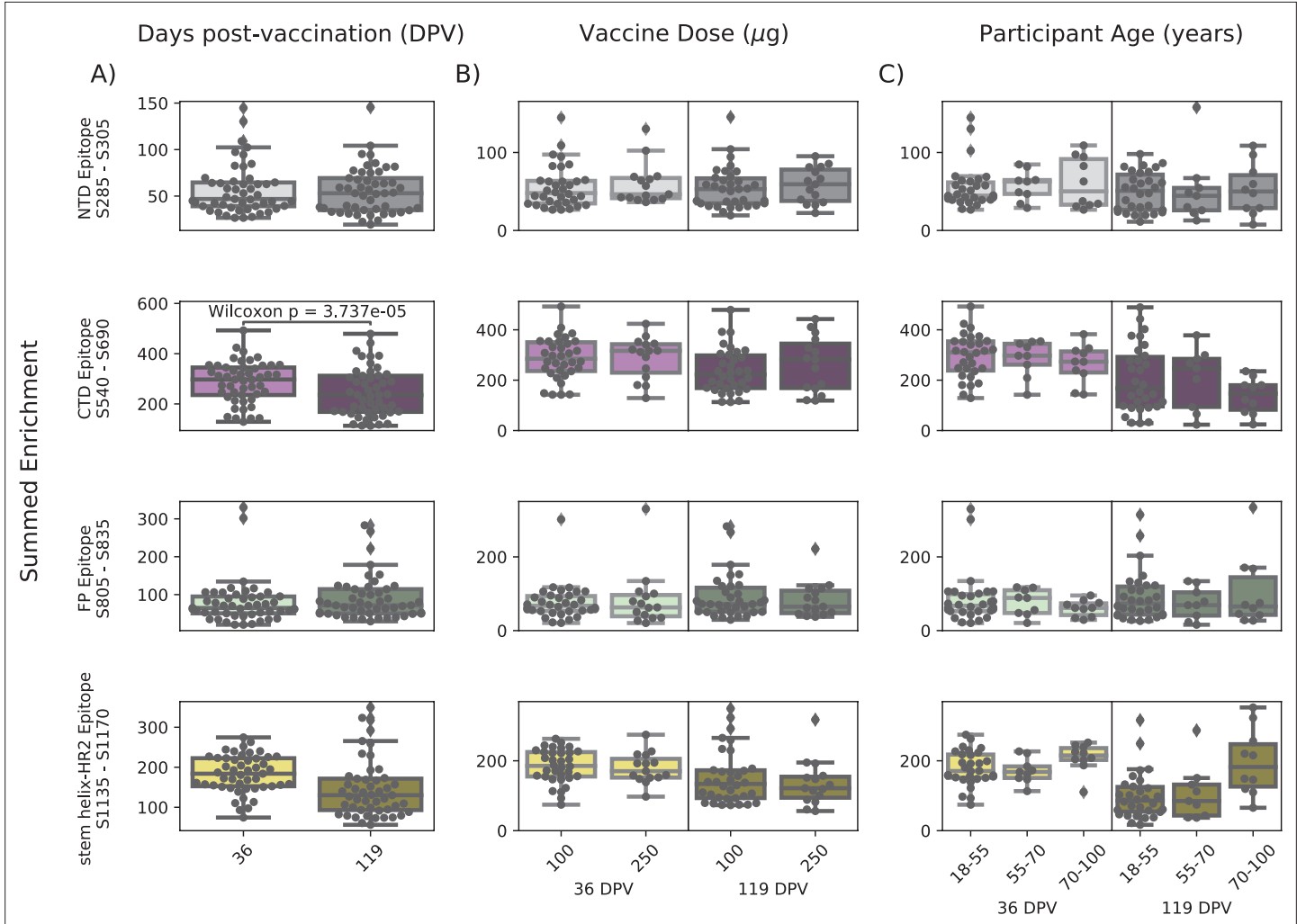

**Figure 3.** Comparison of epitope binding for NIH Moderna Trial subgroups. Boxplots of summed wild-type enrichment within epitope binding regions for samples grouped by (**A**) timepoint post vaccination, (**B**) vaccine dose, or (**C**) participant age. Samples were taken either at 36 (n = 64) or 119 (n = 64) days post vaccination. (**B**) and (**C**) are additionally separated by timepoint post vaccination. Results of a Wilcoxon rank-sum test between the groups appear only where p<0.05 after Bonferroni multiple testing correction (36 group comparisons). Figures containing all p-values for both replicate batches are available at https://github.com/matsengrp/phage-dms-vacc-analysis (swh:1:rev:d4c770ad49ed2f8ab31e499265dd02273cff6f86, **Matsen, 2022**).

The online version of this article includes the following figure supplement(s) for figure 3:

**Figure supplement 1.** Comparison of epitope binding for Hospitalized or Ambulatory Adults with Respiratory Viral Infections (HAARVI) subgroups.

examined as a variable; there appeared to be a difference in epitope binding in the SH-H region, but this did not survive multiple testing correction (*Figure 3C*).

In infected individuals, the effect of time post symptom onset on epitope binding was examined using nonhospitalized infected individuals in the HAARVI Cohort, who were sampled between 26 and 309 days post symptom onset (*Figure 3—figure supplement 1*). Samples were binned into three groups: 0–60, 60–180, and 180–360 days post symptom onset. At all times post symptom onset, there was no significant difference in binding to the four identified epitopes (p>0.05, MWW). Individuals in the HAARVI Cohort were given either the Moderna mRNA-1273 or Pfizer/BioNTech BNT162b2 mRNA vaccine, and comparison of the epitope binding response between the two vaccine types revealed no significant differences in all epitope regions (*Figure 3—figure supplement 1B*, p>0.05, MWW).

## Infection and vaccination shape pathways of escape

The Spike Phage-DMS library contains peptides with every possible single amino acid substitution in addition to the wild-type sequence, enabling us to assay the impact of mutations on antibody binding.

The effect of site-specific substitutions in critical antibody binding regions not only provides a high-resolution picture of the likely epitope intervals, but also identifies mutations that confer escape within the binding region. The effect of each mutation on serum antibody binding was quantified by calculating its scaled differential selection value, a metric that reports log-fold change of mutant peptide binding over wild-type peptide binding at any given site (see 'Materials and methods') (*Garrett et al., 2020*). Site mutations that cause a loss of binding when compared to the wild-type peptide centered at that same site are reported as having negative differential selection values, whereas those that bind better than the wild-type peptide have positive differential selection values. In order for differential selection to be meaningful, however, we must ensure that we do not include weak or sporadic signals that may be due to nonspecific binding. Accordingly, we set a threshold of summed wild-type peptide binding in any one region. By doing so, we lose samples in the analysis but can be confident in the results presented by samples passing this curation step (*Figure 4—figure supplement 1*). For samples that passed this threshold, we compared the effect of prior infection and/or time post vaccination on the pathways of escape in each epitope region as follows. Plots depicting the effect of mutations for all samples are publicly explorable at https://github.com/matsengrp/vacc-dms-view-host-repo (copy archived at swh:1:rev:6519940a17ea2489f445b897485e621d8c6b781d, *Galloway, 2022a*).

## N-terminal domain (NTD) and C-terminal domain (CTD)

We examined the sites of escape within the NTD and CTD epitope regions, focusing on vaccinated individuals from the Moderna Trial Cohort because these epitopes were notable targets of the vaccine response and not commonly found in infected individuals. Vaccination elicited antibodies with a strikingly uniform escape profile in the NTD epitope across samples (*Figure 4A*), with the majority of samples being sensitive to mutation at sites 291, 294–297, 300–302, and 304, which are in the very C-terminal portion of NTD as well as the region between NTD and RBD. The CTD region appeared to consist of multiple epitopes, the dominant being located at the N-terminal region between positions 545–580 (termed CTD-N). Antibodies that bound to this dominant CTD epitope had a less uniform escape profile, but sites 561 and 562 were common sites of escape in most samples (*Figure 4B*). For antibodies to both the NTD and CTD-N epitopes, the pathways of escape tended to drift over time and were different at 119 days post vaccination as compared to 36 days post vaccination.

## Fusion peptide (FP)

Antibodies against the FP epitope are strongly stimulated after infection but are less strongly induced after subsequent vaccination (*Figure 2*). Thus, we investigated whether the pathways of escape for serum antibodies also changed after vaccination within samples from previously infected individuals in the HAARVI Cohort. The escape profiles of antibodies in paired samples that strongly bound to the FP epitope both after infection and after subsequent vaccination are shown as a logo plot (*Figure 5A*). The major sites of escape within the FP epitope for these samples were sites 819, 820, 822, and 823, and these sites of escape did not appear to change after vaccination, although we noted that there was more variability in the escape profiles after vaccination.

We next examined the pathways of escape for FP binding antibodies in vaccinated individuals from the Moderna Trial Cohort. In people with no prior infection, vaccination induced diverse pathways of escape in the FP region (*Figure 5B*). For example, for participant M10 escape was focused on sites 814, 816, and 818, whereas for participant M38 escape was focused on 819, 820, and 823. There appeared to be some differences in the escape profile at 119 days as compared to 36 days post vaccination, as exemplified by participants M15, M17, and M20. However, in general many of the major sites of escape were shared at both timepoints within each individual and as a group.

## Stem helix-heptad repeat 2 (SH-H)

In order to determine the effect of prior infection on the binding profiles of antibodies after vaccination within the SH-H epitope region, we explored the pathways of escape for paired samples from patients with prior infection in the HAARVI Cohort before and after vaccination as was done for the FP region. Samples from previously infected individuals with no vaccination history had diverse pathways of escape within the SH-H epitope (*Figure 6A*). For example, site 1149 was only sensitive to mutation for participant 217C, and site 1157 was only sensitive to mutations for participants 120C and 146C. In contrast, the samples from vaccinated individuals, regardless of infection history, tended to have a

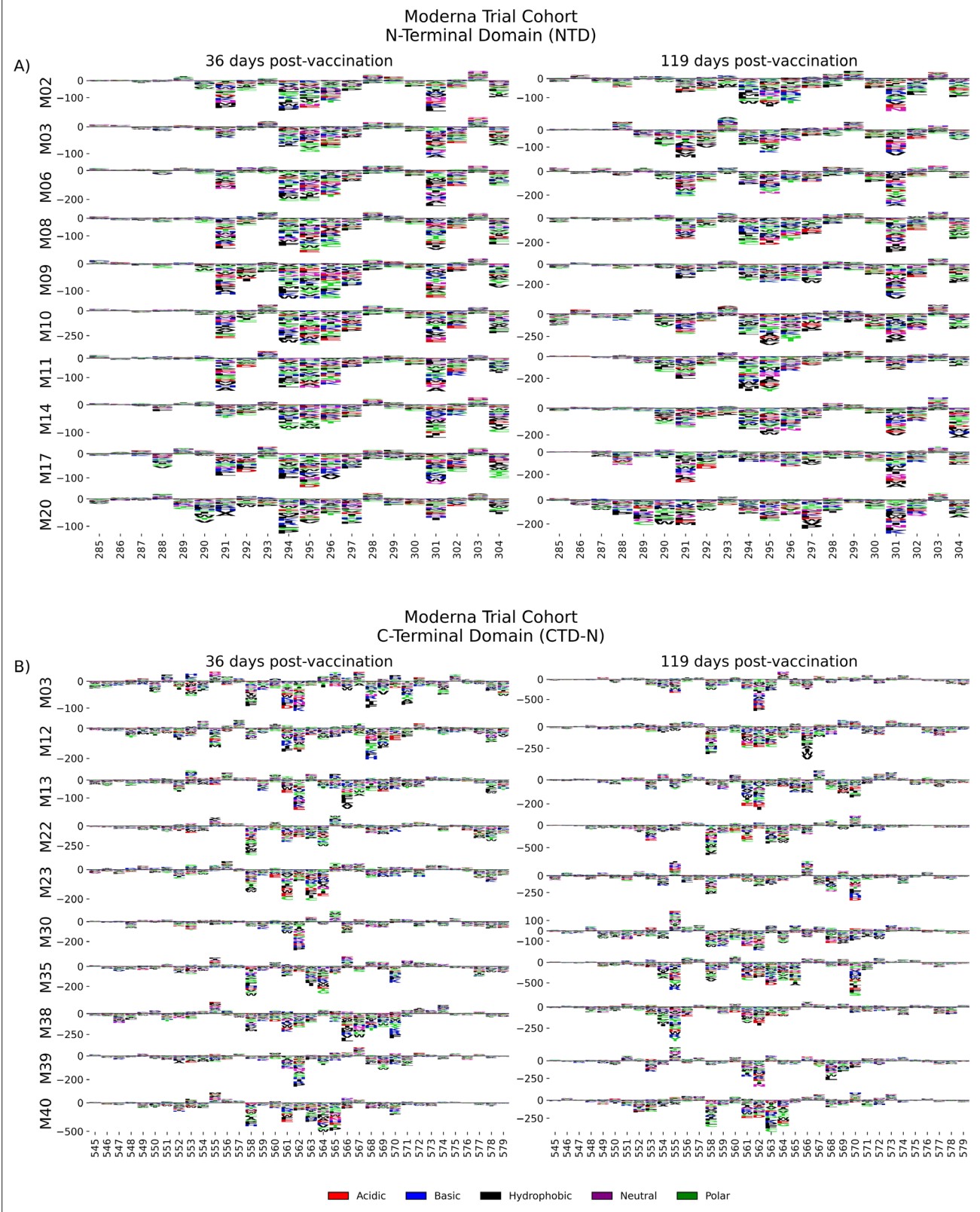

**Figure 4.** NTD and CTD-N epitope escape profiles. (**A, B**) Logo plots depicting the effect of mutations on epitope binding in either the NTD (**A**) or CTD-N (**B**) epitope for paired samples from the Moderna Trial Cohort. The height of the letters corresponds to the magnitude of the effect of that mutation on epitope binding, that is, its scaled differential selection value. Letters below zero indicate mutations that cause poorer antibody binding as compared to wild-type peptide, and letters above zero indicate mutations that bind better than the wild-type peptide. Letter colors denote the

*Figure 4 continued on next page*

*Figure 4 continued*

chemical property of the amino acids. Logo plots on the left and right are paired samples from the same individual, with the participant ID noted on the left.

The online version of this article includes the following figure supplement(s) for figure 4:

**Figure supplement 1.** Thresholding of total epitope binding within major epitope regions.

uniform pathway of escape. The most prominent and consistent sites of escape for vaccinated individuals, both with and without prior infection, were at sites 1148, 1152, 1155, and 1,156. Of note, the pre-vaccination sample from an individual with prior infection requiring hospitalization (participant 6C) displayed an escape profile highly similar to those from vaccinated individuals, and this escape profile did not change after vaccination.

To see whether the pathways of escape changed over time after vaccination, we visualized the escape mutations within the SH-H epitope for the samples in the Moderna Trial Cohort at 36 and 119 days after the first dose of vaccine (*Figure 6B*). We saw a highly uniform pattern of escape for most samples at days 36 and 119, again with escape mainly occurring at sites 1148, 1152, 1155, and

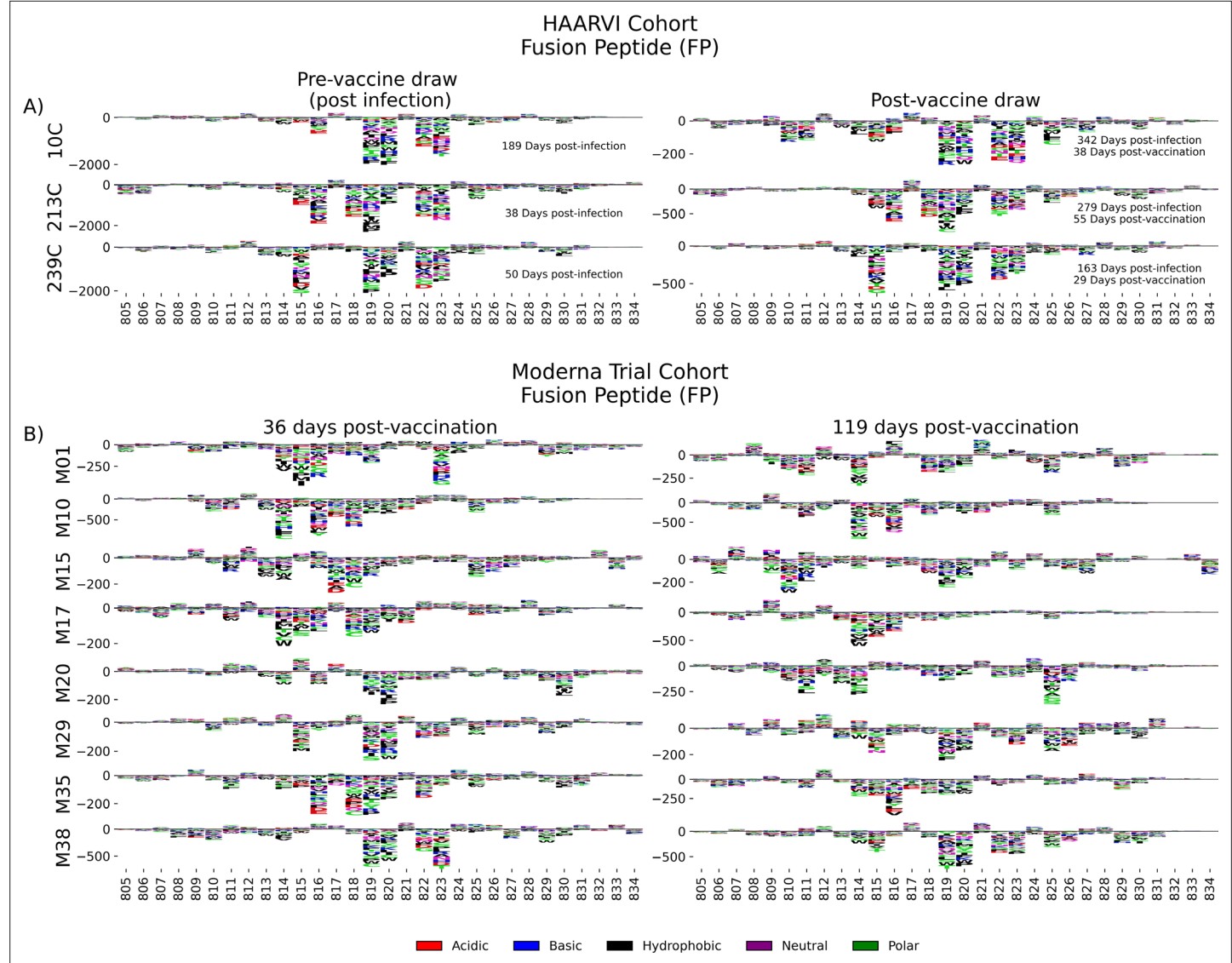

**Figure 5.** Fusion peptide (FP) epitope escape profiles. (**A, B**) Logo plots depicting the effect of mutations on epitope binding within the FP epitope region for paired samples from the (**A**) Hospitalized or Ambulatory Adults with Respiratory Viral Infections (HAARVI) Cohort or (**B**) Moderna Trial Cohort. Details are as described in *Figure 4*.

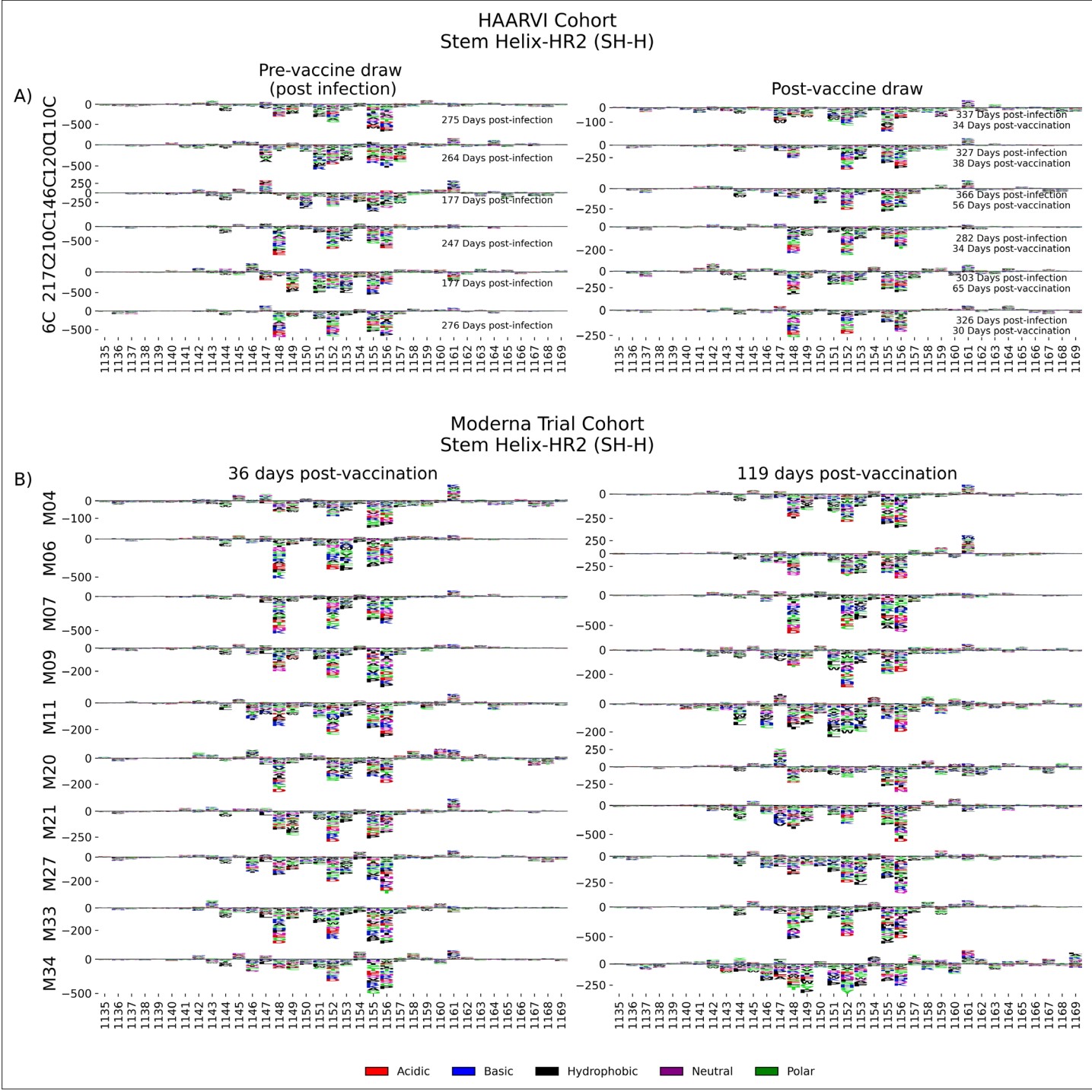

**Figure 6.** SH-H epitope escape profiles. (**A, B**) Logo plots depicting the effect of mutations on epitope binding within the SH-H epitope region for paired samples from the (**A**) Hospitalized or Ambulatory Adults with Respiratory Viral Infections (HAARVI) Cohort or (**B**) Moderna Trial Cohort. Details are as described in *Figure 4*.

1156. For some participants, such as M11, M34, and M35, the escape mutations appeared to drift over time, but the major sites of escape remained the same.

## Discussion

In this study, we comprehensively profiled the antibody response to the SARS-CoV-2 Spike protein, including pathways of escape from sera in individuals with diverse infection and vaccination histories. We identified four major targets of antibody responses outside of the core RBD domains, in the NTD, CTD, FP, and stem helix-HR2 regions. Vaccinated individuals as well as individuals with severe infection requiring hospitalization both had antibodies to these four epitope regions, whereas individuals with mild infection that did not require hospitalization preferentially targeted only FP and SH-H. One explanation for why vaccination and severely infected individuals share this antibody binding profile is that it is a result of the high antigenic load experienced by both groups. In previously infected cases, the epitope binding patterns changed over time after vaccination, with decreased binding to both the CTD and SH-H epitopes. However, there was not uniform decay across all four epitopes, indicating that waning antibody titers may not occur for all epitopes equally. Other factors such as vaccine dose (100 µg or 250 µg), vaccine type (BNT162b2 or mRNA-1273), and participant age did not significantly affect the specificity of the antibody response.

We explored the pathways of escape for antibodies binding to these key regions in infected and vaccinated people. We defined for the first time the escape pathways for NTD and CTD-N binding antibodies, epitopes that were commonly found in vaccinated individuals but not in infected individuals. In the case of the NTD epitope, which was located at the C-terminal end of the NTD, escape mutations were uniform and consistent amongst vaccinees, while pathways of escape were more diverse for CTD-N antibodies. Individuals with antibodies that strongly bound the FP epitope had focused escape profiles, with the majority of escape occurring at sites 819, 820, 822, and 823, although the sample size of this group is small (N = 3). Vaccination did not greatly alter the escape profile in previously infected individuals, nor did vaccination alone induce a strong or uniform response at the FP epitope. In contrast, antibodies that bind in the SH-H epitope region after infection have diverse pathways of escape, while after vaccination they appear to converge on a more uniform pathway of escape that includes mutations at sites 1148, 1152, 1155, and 1156. Interestingly, these are also the sites of contact for a cross-reactive HR2-specific antibody isolated from a mouse sequentially immunized with the MERS and SARS Spike proteins (*Sauer et al., 2021*). This hints that a singular antibody clonotype could be elicited when exposed to a stabilized Spike protein, dominating the response in the SH-H region.

We also observed some drift in the pathways of escape within a single person over time after vaccination. This mirrors findings from a recent study that examined sites of escape for RBD-specific antibodies in serum samples from the same Moderna Trial Cohort as used in this study (*Greaney et al., 2021b*). Together, these results suggest that the B cell response after vaccination with Spike mRNA continues to evolve over time. Multiple studies have demonstrated that SARS-CoV-2-specific B cells undergo continued somatic hypermutation in the months after infection, likely due to antigen persistence (*Gaebler et al., 2021*; *Sakharkar et al., 2021*). Spike antigen has been detected in the lymph nodes at least 3 months after vaccination with BNT162b2, and continued maturation of germinal center B cells could be a possible explanation for the changes in epitope binding we observed (*Turner et al., 2021*). Alternatively, turnover of short-term plasma cells and memory B cells could account for loss of antibody binding to certain epitopes.

Our study has important limitations worth noting. Because the Spike Phage-DMS library displays 31 amino acid peptides, we are unable to detect antibodies that bind to conformational epitopes and/or glycosylated epitopes. This is demonstrated by the lack of observable binding to the RBD region, a domain with complex folding and known target of antibodies from infected and vaccinated individuals. However, prior studies of RBD epitopes have already been reported using an overlapping set of samples from the HAARVI Cohort, and together these results paint a more complete picture of epitopes across the Spike protein (*Greaney et al., 2021a*; *Greaney et al., 2021b*). Finally, we only have five individuals within the hospitalized group and this small sample size limits our ability to make conclusions about epitope binding in those with severe infection.

Our finding that vaccinated individuals have a broader response across the Spike protein than infected individuals may have important implications for immune durability against future

SARS-CoV-2 variants. Evidence suggests that a polyclonal antibody response that is resistant in the face of multiple mutations is necessary for long-lasting immunity against a mutating viral pathogen (*Greaney et al., 2021c*). Thus, the polyclonal response to vaccination may provide greater protection from infection than the more focused response after infection. However, the number of epitopes targeted provides just one benchmark and the ability to escape at the population level could also be influenced by the diversity of individuals' antibody responses at each epitope and thus the likelihood that a single escape mutation could be widely selected. At one S2-domain epitope region (SH-H), vaccination induced uniform sites of escape that may be due to a singular type of antibody that would allow escape by the same mutations for all vaccinated people. However, epitopes in the S2 domain tend to be in highly conserved regions with important functions that constrain the virus' ability to mutate, making escape from these antibodies less likely than for RBD, where escape is already common. Indeed, mutations in the FP and SH-H epitopes are not arising in the global population of SARS-CoV-2 (*Garrett et al., 2021*), providing some suggestion that these regions may be constrained (*Walls et al., 2020*; *Jaroszewski et al., 2021*). Overall, further studies of the functional capacity of these vaccine-elicited antibodies targeting epitopes outside of RBD are warranted to provide a path towards a polyclonal response to epitopes across the full Spike potein. This comprehensive view may further the goal of a more universal coronavirus vaccine that elimin/ates the need for continual updates of the SARS-CoV-2 vaccine strain due to mutations in variable regions on Spike.

# Materials and methods

**Key resources table**

| Reagent type (species) or resource | Designation | Source or reference | Identifiers | Additional information |
|---|---|---|---|---|
| Other | Protein A Dynabeads | Invitrogen | 10001D | |
| Other | Protein G Dynabeads | Invitrogen | 10003D | |
| Commercial assay or kit | Q5 High-Fidelity 2X Master Mix | NEB | M0492L | |
| Commercial assay or kit | AMPure XP beads | Beckman Coulter | A63881 | |
| Commercial assay or kit | Quant-iT PicoGreen dsDNA Assay Kit | Invitrogen | P11496 | |
| Commercial assay or kit | KAPA Library Quantification Kit | Roche | KK4824 | |

## Sample collection
### Moderna Trial Cohort
We obtained post-vaccination serum samples via the National Institute of Allergy and Infection Disease that were taken as part of a phase I clinical trial testing the safety and efficacy of the Moderna mRNA-1273 vaccine (NCT04283461) (*Jackson et al., 2020*). All samples were de-identified, and thus all work was approved by the Fred Hutchinson Cancer Research Center Institutional Review Board as nonhuman subjects research. Trial participants were given either 100 µg or 250 µg doses of the mRNA-1273 vaccine, and serum was sampled from all trial participants at 36 days and 119 days post vaccination. See *Supplementary file 1* for detailed metadata related to each participant and serum sample.

### HAARVI Cohort
We obtained plasma samples from individuals enrolled in the HAARVI study conducted in Seattle (*Garrett et al., 2021*). Individuals were either enrolled upon PCR-confirmed diagnosis with SARS-CoV-2 infection or as control subjects prior to receiving vaccination with either BNT162b2 (Pfizer/BioNTech) or mRNA-1273 (Moderna). See *Supplementary file 1* for detailed metadata related to each participant and plasma sample. For convenience, all plasma and serum samples in this study are referred to as serum. This study was approved by the University of Washington Institutional Review Board.

## Spike Phage-DMS assay

The Spike Phage-DMS library used in this study contained 24,820 designed peptides that tile across the length of the Spike protein. Peptides are each 31 amino acids long and tile by one amino acid increments, and correspond to either the wild-type sequence or a sequence containing a single mutation. Serum samples were profiled using the Spike Phage-DMS library as previously described (*Garrett et al., 2021*). Following this method, the Spike Phage-DMS library was diluted in Phage Extraction Buffer (20 mM Tris-HCl, pH 8.0, 100 mM NaCl, 6 mM MgSO$_4$) to a concentration of 2.964 × 10$^9$ plaque-forming units/mL, which corresponds to approximately 200,000-fold coverage of each peptide. 10 µL of serum or plasma was added to 1 mL of the diluted library and incubated in a deep 96-well plate overnight at 4°C on a rotator. 40 µL of a 1:1 mixture of Protein A and Protein G Dynabeads (Invitrogen) were added to each well and then incubated at 4°C for 4 hr on a rotator. Beads bound to the antibody-phage complex were magnetically separated and washed 3× with 400 µL wash buffer (150 mM NaCl, 50 mM Tris-HCl, 0.1% [vol/vol] NP-40, pH 7.5). Beads were resuspended in 40 µL of water and lysed at 95°C for 10 min. The diluted Spike Phage-DMS library was also lysed to capture the starting frequencies of peptides. All samples were run twice, once each with two independently generated Spike Phage-DMS libraries.

DNA from lysed samples were amplified and sequenced as previously described (*Garrett et al., 2021*). Two rounds of PCR were performed using Q5 High-Fidelity 2X Master Mix (NEB). For the first round of PCR, 10 µL of lysed phage was used as the template in a 25 µL reaction using the primers described in *Garrett et al., 2021*. For the second round of PCR, 2 µL of the round 1 PCR product was then used as the template in a 50 µL reaction, with primers that add dual indexing sequences on either side of the insert. PCR products were then cleaned using AMPure XP beads (Beckman Coulter) and eluted in 50 µL water. DNA concentrations were quantified via Quant-iT PicoGreen dsDNA Assay Kit (Invitrogen). Equimolar amounts of DNA from the samples, along with 10× the amount of the input library samples, was pooled, gel purified, and the final library was quantified using the KAPA Library Quantification Kit (Roche). Pools were sequenced on an Illumina HiSeq 2500 machine using the rapid run setting with single end reads.

## Sample curation and replicate structure

All sample IPs and downstream analysis were run in duplicate across two separate phage display library batches to ensure reproducibility, with the exception of the four acute samples from hospitalized HAARVI participants that were run in singlicate. All results were cross-checked with the set of batch replicates to ensure significance fell within one order of magnitude where applicable. For brevity, we present only figures resulting from the single complete set of batch-specific replicates; however, all figures using the second set of library batch replicates are available (see 'Code availability' and 'Data availability'). Additionally, some samples were run with 'in-line' technical replicates within the same batch. In the case with more than one technical replicate, we selected the sample with the highest reads mapped from each set of batch replicates for our downstream analysis.

## Short-read alignment and peptide counts processing

Samples were aliquoted and sequenced targeting 10× coverage of total sample reads to the peptide library reference. We demultiplexed samples using Illumina MiSeq Reporter software. Post sample demultiplexing, we used a *Nextflow* pipeline to process the peptide counts as well as alignment stats for all samples (*Di Tommaso et al., 2017*). The tools and parameters describing the workflow are as follows. The index creation and short-read alignment step were done using *Bowtie2*. During alignment, we allowed for zero mismatches in the default seed length of each read (20, very sensitive) after trimming 32 bases from the 3′ end of each 125 bp read to match the 93 bp peptides in our reference library (*Langmead and Salzberg, 2012*). *Samtools* was subsequently used to gather sequencing statistics as well as produce the final peptide counts using the *stats* and *idxstats* modules. Finally, the pipeline collected all reference peptide alignment counts and merged them into a single *xarray* dataset coupling sample and peptide metadata with their respective count.

Each of the processing steps described here, as well as downstream analysis and plotting, was run using static and freely available Docker containers for reproducibility. We provide an automated workflow and the configuration scripts defining exact parameters. See 'Code availability' and 'Data availability' section for more information.

## Epitope binding region identification

PCA via singular value decomposition was performed on each set of batch replicates using the scikit-learn package (*Pedregosa et al., 2011*). We first subset our dataset to only include wild-type peptide count enrichments from either infected or vaccinated individuals as input. This curation resulted in the matrix, $X$ of size $n \times p$ with $n$ biologically distinct replicates and $p$ enrichment features across the spike protein. All enrichment values were calculated as a fold change in the frequency for any one sample enrichment over the library control enrichment at the same sites. Each feature was mean centered before performing the PCA such that the covariance matrix of $X$ is equivalent to $X^T X / (n-1)$. We can then use the eigendecomposition, $X = USV^T$, to describe the data. The principal axes in feature space are then represented by the columns of $V$ and represent the direction of maximum variance in the data. *Figure 2—figure supplement 1* shows three facets of this decomposition; *Figure 2—figure supplement 1A* shows the unit scaled sample 'scores' represented by the columns to visualize sample relationship in principal component space; *Figure 2—figure supplement 1B* shows the component loadings (scaled by the square root of the respective eigenvalues in S); and *Figure 2—figure supplement 1C* shows the first three principal axes/directions in feature space plotted as a function of the WT peptide feature location on the Spike protein. Together, these provide a visualization of key features in the data used in our downstream analysis. We chose our epitope regions as contiguous regions of nonzero value in the loadings in the first three principal axes.

## Identifying high-resolution pathways of escape

In order to ensure reliable measurements of differential selection of single AA variants compared to the ancestral sequence variant, we threw out samples whose respective sum of wild-type enrichment was below a threshold set for each of the defined binding regions (*Figure 4—figure supplement 1*). Once curated, we computed the log-fold change in each of the 19 possible variant substitutions at each site. This metric was then scaled by the average of the wild-type sequence enrichment coupled with both the preceding and following wild-type peptide enrichments at any given site. To evaluate escape at each site, we then sum the differential selection metric as described for each variant at a site to examine a more complete picture of the data defining escape patterns in each sample group.

## Code availability

We provide a fully reproducible automated workflow that ingests raw sequencing data and performs all analyses presented in the paper. The workflow defines and runs the processing steps within publicly available and static Docker software containers, including *phippery* and *phip-flow* described in the 'Materials and methods' section. The source code, Nextflow script, software dependencies, and instructions for rerunning the analysis can be found at https://github.com/matsengrp/phage-dms-vacc-analysis (copy archived at swh:1:rev:d4c770ad49ed2f8ab31e499265dd02273cff6f86, *Matsen, 2022*).

The generalized PhIP-Seq alignment and count generation pipeline script can be found at https://github.com/matsengrp/phip-flow (copy archived at swh:1:rev:2bffd776688efa13a9953a5fe5cba47a17590578, *Galloway, 2022c*). A template and documentation for the alignment pipeline configuration are available at https://github.com/matsengrp/phip-flow-template (copy archived at swh:1:rev:c35d2c9532ff5ef450783f52f850207de8ad87fb, *Galloway, 2022b*). Finally, we provide a Python API, *phippery,* to query the resulting dataset post alignment that can be found at https://github.com/matsengrp/phippery (copy archived at swh:1:rev:871c65fe331ebabbb48338f1d9a-0d518a399cc78; *Galloway, 2022d*).

All raw sequencing data was submitted to the NCBI SRA under PRJNA765705. Preprocessed enrichment data is available upon request. Additionally, differential selection data and more can be explored interactively using the dms-view toolkit available at https://github.com/matsengrp/vacc-dms-view-host-repo (copy archived at swh:1:rev:6519940a17ea2489f445b897485e621d8c6b781d, *Galloway, 2022a*).

For more information regarding code and data availability, please email jgallowa@fredhutch.org. For original data from the NIH Moderna trial, please see *Jackson et al., 2020*; and for information on the HAARVI Cohort, please contact HYC.

## Statistical analysis

Estimates of significance presented between group continuous distributions of wild-type enrichment were reported using a Mann–Whitney–Wilcoxon test with the exception of analysis that included only paired longitudinal samples – such as the comparison of 36 and 119 days post vaccination – in this case, we used a Wilcoxon signed-rank test. Bonferroni correction was applied where applicable, and adjusted p-values 0.05 were presented as significant. All statistical analyses were done using Python 3.6 and plotted using the *statannot* package found at https://github.com/webermarcolivier/statannot, *Weber, 2022*. The static Docker container used for all statistical analyses is publicly hosted at https://quay.io/repository/matsengrp/vacc-ms-analysis.

# Acknowledgements

We thank Kevin Sung, Thayer Fisher, and Noah Simon for providing advice regarding computational analyses. We thank Cassie Sather and the Genomics core facility for assistance with sequencing. We thank Laura Jackson (Kaiser Permanente), Chris Roberts, Catherine Luke, and Rebecca Lampley (National Institute of Allergy and Infectious Diseases [NIAID], NIH) for assistance with obtaining the mRNA-1273 phase I trial vaccine samples. We also thank all research participants and study staff of the Hospitalized or Ambulatory Adults with Respiratory Viral Infections (HAARVI) study. This work was supported by NIH grants AI138709 (PI JMO) and AI146028 (PI FAM). JMO received support as the Endowed Chair for Graduate Education (FHCRC). The research of FAM was supported in part by a Faculty Scholar grant from the Howard Hughes Medical Institute and the Simons Foundation. Scientific Computing Infrastructure at Fred Hutch was funded by ORIP grant S10OD028685.

# Additional information

### Competing interests

Helen Y Chu: reported consulting with Ellume, Pfizer, The Bill and Melinda Gates Foundation, Glaxo Smith Kline, and Merck. She has received research funding from Gates Ventures, Sanofi Pasteur, and support and reagents from Ellume and Cepheid outside of the submitted work. Julie M Overbaugh: Reviewing editor, *eLife*. The other authors declare that no competing interests exist.

### Funding

| Funder | Grant reference number | Author |
| --- | --- | --- |
| National Institutes of Health | 5R01AI138709-04 | Julie M Overbaugh |
| National Institutes of Health | 1R01AI146028-01 | Frederick A Matsen |

The funders had no role in study design, data collection and interpretation, or the decision to submit the work for publication.

### Author contributions

Meghan E Garrett, Data curation, Formal analysis, Investigation, Methodology, Validation, Visualization, Writing - original draft; Jared G Galloway, Data curation, Formal analysis, Methodology, Software, Validation, Visualization, Writing - original draft; Caitlin Wolf, Jennifer K Logue, Nicholas Franko, Resources; Helen Y Chu, Funding acquisition, Resources, Supervision, Writing – review and editing; Frederick A Matsen, Conceptualization, Funding acquisition, Methodology, Project administration, Resources, Software, Supervision, Writing – review and editing; Julie M Overbaugh, Conceptualization, Funding acquisition, Methodology, Project administration, Supervision, Writing – review and editing

### Author ORCIDs

Nicholas Franko http://orcid.org/0000-0001-8165-6332
Frederick A Matsen IV, http://orcid.org/0000-0003-0607-6025
Julie M Overbaugh http://orcid.org/0000-0002-0239-9444

### Ethics

Clinical trial registration NCT04283461.

Human subjects: Moderna Trial Cohort We obtained post-vaccination serum samples via the National Institute of Allergy and Infection Disease that were taken as part of a phase I clinical trial testing the safety and efficacy of the Moderna mRNA-1273 vaccine (NCT04283461). All samples were de-identified and thus all work was approved by the Fred Hutchinson Cancer Research Center Institutional Review Board as nonhuman subjects research. HAARVI Cohort We obtained plasma samples from individuals enrolled in the Hospitalized or Ambulatory Adults with Respiratory Viral Infections (HAARVI) study conducted in Seattle. Individuals were either enrolled upon PCR confirmed diagnosis with SARS-CoV-2 infection or as control subjects prior to receiving vaccination with either BNT162b2 (Pfizer/BioNTech) or mRNA-1273 (Moderna). Electronic informed consent was obtained for every individual, and the study was approved by the University of Washington Institutional Review Board.

### Decision letter and Author response

Decision letter https://doi.org/10.7554/eLife.73490.sa1
Author response https://doi.org/10.7554/eLife.73490.sa2

---

## Additional files

### Supplementary files

- Transparent reporting form
- Supplementary file 1. Cohort information.

### Data availability

We provide a fully reproducible automated workflow which ingests raw sequencing data and performs all analyses presented in the paper. The workflow defines and runs the processing steps within publicly available and static Docker software containers, including phippery and phip-flow described in the Methods section. The source code, Nextflow script, software dependencies, and instructions for re-running the analysis can be found at (https://github.com/matsengrp/phage-dms-vacc-analysis copy archived at swh:1:rev:d4c770ad49ed2f8ab31e499265dd02273cff6f86). The generalized PhIP-Seq alignment and count generation pipeline script can be found at (https://github.com/matsengrp/phip-flow copy archived at swh:1:rev:2bffd776688efa13a9953a5fe5cba47a17590578). A template and documentation for the alignment pipeline configuration is available at (https://github.com/matsengrp/phip-flow-template copy archived at swh:1:rev:c35d2c9532ff5ef450783f52f850207de8ad87fb). Finally, we provide a python API, phippery, to query the resulting dataset post-alignment that can be found at (https://github.com/matsengrp/phippery copy archived at swh:1:rev:871c65fe331ebabbb48338f-1d9a0d518a399cc78). All raw sequencing data was submitted to the NCBI SRA under PRJNA765705. Pre-processed enrichment data is available upon request. Additionally, differential selection data and more can be explored interactively using the dms-view toolkit available at (https://github.com/matsengrp/vacc-dms-view-host-repo copy archived at swh:1:rev:6519940a17ea2489f445b897485e621d8c6b781d). For more information regarding code and data availability, please email jgallowa@fredhutch.org. For original data from the NIH Moderna trial please see Jackson et al32, and for information on the HAARVI cohort please contact HYC. A template and documentation for the alignment pipeline configuration is available at (https://github.com/matsengrp/phip-flow-template). Finally, we provide a python API, phippery, to query the resulting dataset post-alignment that can be found at (https://github.com/matsengrp/phippery). All raw sequencing data was submitted to the NCBI SRA. Pre-processed enrichment data is available upon request. Additionally, differential selection data and more can be explored interactively using the dms-view toolkit available at (https://github.com/matsengrp/vacc-dms-view-host-repo). For more information, please email jgallowa@fredhutch.org.

The following dataset was generated:

| Author(s) | Year | Dataset title | Dataset URL | Database and Identifier |
|---|---|---|---|---|
| Garrett ME, Galloway JG, Wolf C, Logue JK, Franko N, Chu HY, Matsen IV FA, Overbaugh J | 2021 | Phage-DMS with serum/plasma from SARS-CoV-2 infected and vaccinated individuals | https://www.ncbi.nlm.nih.gov/bioproject/765705 | NCBI BioProject, PRJNA765705 |

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
