## [Editor Report]

In this paper, antibody responses to SARS-CoV-2 Spike protein were investigated in mRNA vaccinated, SARS-CoV-2 infected with mild or severe COVID-19 and vaccinated individuals with prior infection. Key findings of the study are that severe COVID-19 and vaccinated individuals had higher binding to Spike protein regions NTD, CTD, and SH-H compared to individuals with mild COVID-19; while mild COVID-19 infections had higher binding to FP than vaccinated or severe COVID-19 individuals. Individuals with or without prior infection were not different and that covariates did not appear to impact the antibody recognition profiles. The authors identified potential escape pathways in these epitope regions, some of which differed between vaccination and infection or drifted over time. The authors acknowledge that this approach is limited to linear epitopes and does not include receptor-binding domain epitopes. The study provides novel insight into the major epitope regions targeted by polyclonal antibodies elicited by vaccination vs. infection, as well as potential pathways for the virus to escape recognition.

---

## [Decision Letter]

**Decision letter after peer review:**

Thank you for submitting your article "Comprehensive characterization of the antibody responses to SARS-CoV-2 Spike protein after infection and/or vaccination" for consideration by *eLife*. Your article has been reviewed by 2 peer reviewers, and the evaluation has been overseen by a Reviewing Editor and Jos Van der Meer as the Senior Editor. The following individual involved in review of your submission has agreed to reveal their identity: Bonnie Gunn (Reviewer #2).

Essential revisions:

1. We are curious about the potential decay in CTD-binding antibodies, especially given that non-hospitalized infection-induced antibodies show limited binding to CTD and the reduced levels of CTD binding in vaccinees at D119 compared with D36 post-vaccination. Is it possible that humoral immunity against CTD may be an early signature that decays and may be undetectable at later convalescent time points in infected individuals? The data presented in Supplemental Figure 2 shows that a subset of people between make CTD antibodies. Is there longitudinal data (or patients within the HAARVI cohort that the authors have analyzed) from acute or early convalescence to determine if CTD binding antibodies persist or decline? Alternatively, as the hospitalized individuals also developed responses against CTD, could the development of these antibodies be dependent on level of antigen load (either in infection or through vaccination)?

2. The finding that non-hospitalized infected individuals have elevated binding to the FP peptide compared with the other groups is an interesting finding that may contribute our understanding of immune responses that prevent hospitalization. Do these individuals have elevated neutralizing activity if the amount of spike-binding antibodies is normalized?

3. Is there high sequence similarity in the FP epitope between the HCoVs and SARS-CoV-2 that could explain the high levels of binding in the naïve sera?

---

## [Author Response]

Essential revisions:1. We are curious about the potential decay in CTD-binding antibodies, especially given that non-hospitalized infection-induced antibodies show limited binding to CTD and the reduced levels of CTD binding in vaccinees at D119 compared with D36 post-vaccination. Is it possible that humoral immunity against CTD may be an early signature that decays and may be undetectable at later convalescent time points in infected individuals? The data presented in Supplemental Figure 2 shows that a subset of people between make CTD antibodies. Is there longitudinal data (or patients within the HAARVI cohort that the authors have analyzed) from acute or early convalescence to determine if CTD binding antibodies persist or decline? Alternatively, as the hospitalized individuals also developed responses against CTD, could the development of these antibodies be dependent on level of antigen load (either in infection or through vaccination)?

The reviewers have brought up an interesting point that is consistent with our hypothesis that the higher antigen load experienced by hospitalized or vaccinated individuals why CTD antibodies are similarly high in these groups. We have added a sentence to the discussion that more clearly states this hypothesis for the reader (lines 308-309):

“One explanation for why vaccination and severely infected individuals share this antibody binding profile is that it is a result of the high antigenic load experienced by both groups.”

However, we unfortunately do not have longitudinal data that can be used to directly test this hypothesis. Among the cases studied here, we only have longitudinal samples from one hospitalized individual and do not want to draw on an N of 1 case study to try to comment on whether the CTD antibodies persist or decline in this group. A larger sample set from hospitalized individuals would be needed to address this question.

2. The finding that non-hospitalized infected individuals have elevated binding to the FP peptide compared with the other groups is an interesting finding that may contribute our understanding of immune responses that prevent hospitalization. Do these individuals have elevated neutralizing activity if the amount of spike-binding antibodies is normalized?

It would indeed be exciting if the presence of FP antibodies was associated with clinical outcome, which is a question that this study will hopefully prompt. In this regard, it will also be interesting to determine if the basis for their activity is neutralization and/or other antibody activities. Unfortunately, we do not have data on neutralization that we can use to examine the association suggested by the reviewer and it would take some time to generate such data from this large sample set, which we believe is outside of the scope of this report.

3. Is there high sequence similarity in the FP epitope between the HCoVs and SARS-CoV-2 that could explain the high levels of binding in the naïve sera?

Indeed the sequence similarity in the FP epitope is higher than all other regions on the Spike protein and likely explains the reactivity we see in naive sera. Lines 177-179 (beginning with “These findings likely reflect…”) address this point.